# OpenReview forum: "Dual-Space Smoothness for Robust and Balanced LLM Unlearning"
_ICLR.cc/2026/Conference — ICLR 2026 Poster_

### Official Review · Reviewer_KUGw · 2025-10-30

**Soundness:** 2
**Presentation:** 1
**Contribution:** 2
**Rating:** 2
**Confidence:** 4

**Summary:**

The paper proposes PRISM, a unified "dual-space smoothness" unlearning framework that couples representation-space robustness (via a trained probe) with parameter-space smoothing (SAM-style), explicitly targeting both jailbreak (Prefill, AutoDAN, Multi-turn) and relearning attacks (retrain unlearned models on forget-samples).

**Strengths:**

+ Method engineering: Integrates three components: (1) robust probe guidance, (2) weight-space smoothness, and (3) gradient orthogonalization, into a min-max training loop. The components are modular and reusable in other unlearning pipelines.
+ Evaluates across two distinct scenarios, including continuous text and conversational dialog, and two model families. Covers multiple attack types (prefill, AutoDAN, multi-turn) and relearning scenarios. Includes an ablation study on representation smoothness, parameter smoothness, and gradient decoupling, along with sensitivity analyses on hyperparameters and probe layer selection.

**Weaknesses:**

## Major Issues
**Motivation, notation, and threat model:**
1. *Superficial motivation*: The paper shows GA and SAM+NPO failing with utility collapse, but does not investigate why these methods fail or whether simpler fixes (e.g., with retain loss or regularization) could address the issues. The collapse may represent *poor hyperparameter choices rather than fundamental method limitations*.
2. *Severe mathematical notation and formulation issues*:
+ Undefined notation: For example, lines 135–141, the paper states that Let $z(x):= g(h(x))$ denote the PCA-reduced representation,” and  "Define the acceptance center $c_{a} := \frac{1}{|\mathcal{D}_{a}|} \sum_{x \in \mathcal{D}_a} z(x)$...".

However, the functions $g$, $h$, and dataset $\mathcal{D}_a$ are not formally defined.  The dimensionality of $g$ and $h$ is also missing.

+ Inconsistent notation: For example, lines 211-212 (Section 4.1): "$z(x):= h_{\theta_{0},L}(x) \in \mathbb{R}^{d} ....$" This contradicts the earlier. In Eqn. 8, $\mathcal{L}_{\text{gen}}$ is undefined. What is this loss component? I spotted several similar issues across the paper; I will not correct everything. This is a major problem for a paper at this venue, as it makes the paper hard to understand and less clear.
3. *One more critical point is that the unlearning problem formulation and threat model are missing*: The paper discusses jailbreak and relearning attacks, but lacks a formal threat model: What does the adversary know about the victim's architecture, data, or training procedure? Can the adversary query the model during attacks? What are the budget and computational resources constraints of the adversary? Defining a clear threat model and problem formulation provides a clearer picture of when the method should or should not be expected to hold.

To my taste, *the motivation is weak, and the presentation is poor*.

**Methodology**:

1. The idea of the paper is motivated by SAM, and SAM is well established for machine unlearning [1]. Perturbing latent space to defend has been explored for a long time [2]. The idea of gradient projection/orthogonalization so that the forget gradient is orthogonal to the retain gradients is well-established in machine unlearning, such as [3],[4],[5],[6]. PRISM appears to recombine these ideas into one. Method originality is weak, in my opinion.
2. PRISM is a min–max optimization based unlearning framework aimed to improve robustness, balance forgetting, utility, and stability. However, the paper did not discuss the **theoretical analysis of convergence properties or optimality guarantees for the min-max formulation**.
3. All the analysis in the paper uses the first-order Taylor approximation. First, this assumption used for approximations should be stated more clearly. Importantly, I am not convinced that this approximation is good when the *added perturbation is relatively large*. Deep networks are highly nonlinear, so I wonder if there is a way to justify this assumption better. An empirical analysis of the approximation error would make the approximation more convincing.
4. The paper claims to enforce "dual-space smoothness," but the mathematical connection between representation space and parameter space smoothness is not rigorously established. Smoothness in each space has been shown to improve model robustness. Why smoothness in both spaces should improve robustness?
5. Probe-guided forgetting seems like we know exactly attacker is, then use the attacker's knowledge to develop the defender; this scenario seems impractical.
6. There is a well-known phenomenon in adversarial machine learning that, when defending against a specific kind of attack, might cause an increase in vulnerability to other attacks [7].  Not investigating whether PRISM introduces new vulnerabilities is a critical omission.

**Experiment**

1. PRISM uses two-stage training (probe + unlearning), other baselines use single-stage training. I wonder whether this comparison is fair. In a continuous unlearning setting, where unlearning requests arrive over time, how would PRISM be applied? Would it require retaining the probe?
2. Some important baselines are missing, such as Representation Misdirection for Unlearning [10].
3. The two-stage training (probe + unlearning) significantly increases computational overhead, for example, in Table 1, PRISM: 11.223 seconds/step vs. NPO: 6.475 seconds/step (~73% overhead).
4. Comparisons to other defense methods are missing, making it difficult to assess PRISM's effectiveness.
5. Missing recent works on robust unlearning, such as [8][9].
6. Relearning attacks use "randomly sampled small subsets" (Appendix line 1539), yet the exact subset sizes are not specified, and no analysis is provided on the variability across different random samples.
7. Layer hyperparameter sensitivity: PRISM depends on selecting a specific intermediate layer to extract the representation. The probe's effectiveness is highly sensitive to layer choice, as different layers encode different types of information. Is there a principal way to find the optimal layer?
8. Experiments primarily on 7B-8B models; scalability to larger models unclear.  It would be nice if the paper discusses PRISM's performance on larger models and other architectures, such as Mixture-of-Experts.

## Minor issues
+ Line 159, line 162:  Figure 3b/a not Figure 2b/a
+ Line 222-223: notation error.

I suggest the authors should carefully proofread their paper before updating the script.

## References
[1] Fan, Chongyu, et al. "Towards LLM Unlearning Resilient to Relearning Attacks: A Sharpness-Aware Minimization Perspective and Beyond." Forty-second International Conference on Machine Learning.

[2] He, Zhezhi, Adnan Siraj Rakin, and Deliang Fan. "Parametric noise injection: Trainable randomness to improve deep neural network robustness against adversarial attack." Proceedings of the IEEE/CVF conference on computer vision and pattern recognition. 2019.

[3] Shamsian, Aviv, et al. "Go Beyond Your Means: Unlearning with Per-Sample Gradient Orthogonalization." arXiv preprint arXiv:2503.02312 (2025).

[4] Pan, Zibin, et al. "Federated unlearning with gradient descent and conflict mitigation." Proceedings of the AAAI Conference on Artificial Intelligence. Vol. 39. No. 19. 2025.

[5] Hoang, Tuan, et al. "Learn to unlearn for deep neural networks: Minimizing unlearning interference with gradient projection." Proceedings of the IEEE/CVF Winter Conference on Applications of Computer Vision. 2024.

[6] Wang, Yue, et al. "GRU: Mitigating the Trade-off between Unlearning and Retention for LLMs." arXiv preprint arXiv:2503.09117 (2025).

[7] Tramer, Florian, and Dan Boneh. "Adversarial training and robustness for multiple perturbations." Advances in neural information processing systems 32 (2019).

[8] Łucki, Jakub, et al. "An Adversarial Perspective on Machine Unlearning for AI Safety." Transactions on Machine Learning Research.

[9] Thaker, Pratiksha, et al. "Position: Llm unlearning benchmarks are weak measures of progress." 2025 IEEE Conference on Secure and Trustworthy Machine Learning (SaTML). IEEE, 2025.

[10] Li, Nathaniel, et al. "The WMDP Benchmark: Measuring and Reducing Malicious Use with Unlearning." International Conference on Machine Learning. PMLR, 2024.

**Questions:**

+ How does the quality of the probe affect unlearning performance and robustness?
+ The WMDP benchmark consists of 2 forget sets (WMDP biology and WMDP cyber) alongside 2 retain sets corresponding (can use Wikitext instead, as in [10]). Regarding Section 5.1, line 314-315, the paper states that "the forget set is composed of questions randomly sample from the paraphrased WMDP-bio". What is the "paraphrased" WMDP-bio? Why do we need the retain data generated by the unlearned models with WMDP queries?
+ Please see Major Issues

---

> ### Author Response · Authors · 2025-11-24
>
> We appreciate Reviewer KUGw’s careful evaluation of our work. The constructive criticism and insightful questions help us further improve the paper. We respond to each key question below. We also add additional experiments' results through **anonymous** links for your reference.
>
> ### Motivation, notation, and threat model:
>
>  **W1: Response to Motivation's Limitations**
> We thank the reviewer for this comment. We agree that poor hyperparameters can hurt baseline performance in general, but in our case, the failures of GA and SAM+NPO go beyond tuning issues. Prior work has already shown that GA-style unlearning damages utility severely, even with regularization both theoretically and empirically (e.g., NPO [1], MUSE [2]), and we reproduce these results in Appendix F.3 (Lines 1200–1208) both with and without regularization. For SAM+NPO [3], one of very recent SOTA baselines, we directly evaluate the authors’ released unlearned checkpoints on HuggingFace and still observe clear utility collapse under their tuned setup. Moreover, our motivation is **not only utility collapse**; we also focus on **imbalance** between forget/retain behaviors and **vulnerability** to attacks, where our pilot study shows that existing baselines remain fragile. From these perspectives, we respectfully believe our motivation is substantive rather than superficial.
>
>
> **W2: Response to notation**
> We thank the reviewer for the detailed comments on notation and clarity. The issues raised mainly concern notational completeness, and we appreciate the opportunity to improve the presentation. Our definitions of the jailbreak attacks follow the formulation in [13], but we agree that the notation in the earlier draft could be clearer. In the revised version, we have fully clarified several symbols to avoid ambiguities (Line 149-155).
>
> **W3 & W14: Threat Models**
> We thank the reviewer for raising this important point. We agree that a well-specified threat model is valuable for the robustness of the unlearning method. As discussed in [8], the current unlearning community has not yet placed systematic emphasis on explicit threat-model definitions. We expect that a comprehensive understanding of which attack dimensions are most relevant for evaluating defenses will ultimately require a **broader community effort** over time. In this paper, we take a step in that direction by explicitly stating the threat model we adopt for jailbreak and relearning attacks, following [8][9]. Concretely, we consider the threat model in query distribution and in weight access. For jailbreak attacks, we assume a black-box adversary that can only issue queries to the model (via prompts) but has no access to model weights. For relearning attacks, we assume a white-box adversary with full access to the model parameters. We have clarified the threat models in the revised version to make our problem formulation and evaluation setting more explicit (Line 172-178 and Line 1672-1697).

---

> ### Author Response · Authors · 2025-11-24
>
> ### Method:
> **W4: Response to method originality**:
> We thank the reviewer for this insightful comment. Our work is motivated by SAM [3][5], relearn attacks, jailbreak attacks [13] and latent-space perturbation. In particular, the geometric understanding of how jailbreak and relearning attacks exploit sensitive directions motivates us to use smoothness to address these robustness issues. Our contribution is the novel dual-space smoothness optimization, in both the representation space and the parameter space, to construct probe-based, targeted perturbations that improve robustness against both jailbreak and relearning attacks. To the best of our knowledge, this dual-space smoothness formulation has not been explored in prior unlearning literature. Regarding prior work, methods such as [6] sample random Gaussian noise inside deep networks on images, which is fundamentally different from our setting. Similarly, [7] applies random perturbations to hidden states but have been shown to be vulnerable to relearning attacks, as shown in **[Table. R1](https://ibb.co/qY2YLkgf)**. Existing gradient-projection works largely operate in standard supervised DNN settings [10], per-sample orthogonalization [8,11] and federated unlearning [9] rather than the dual-space optimization setting we study. The core novelty of our method lies in the dual-space smoothness optimization, not in gradient projection itself. We believe this design provides a principled and effective way to achieve robustness, rather than a mechanical combination of existing ideas.
>
>
> **W5, W7: Response to theoretical analysis**:
> During the rebuttal, we try to sketch a formal mathematical argument linking smoothness in representation space and parameter space to robustness and optimality. Adversarial training of the probe can be viewed as imposing a local Lipschitz bound in feature space, which gives a lower bound on the feature perturbation needed for a successful jailbreak. SAM-style smoothing in parameter space imposes a local Lipschitz bound on the forget loss with respect to the weights, which gives a lower bound on the parameter change (or number of steps) required for relearning. By the chain rule, the overall sensitivity of the loss to input or parameter perturbations can then be bounded by the product of these two Lipschitz constants. Reducing both constants, which corresponds to dual-space smoothness, yields a tighter upper bound than smoothing in only one space. As a result, the minimal jailbreak and relearning perturbations become larger, so dual-space smoothness yields stronger robustness. We will make our best effort to complete this proof and include it in the revised version if successful; if not, we will clearly outline it as future work and discuss the associated challenges in detail.
>
> **W6: Response to first-order Taylor approximation**:
> We thank the reviewer for raising this point and we will make the assumption used for our approximations more explicit in the revision. Regarding the first-order Taylor approximation, our design is inspired by Sharpness-Aware Minimization (SAM) [5], which proves that the first-order Taylor approximation to the inner maximization problem to obtain the perturbation is efficient and effective, as implemented in common libraries such as JAX, TensorFlow, and PyTorch. In SAM, the explicit second-order (Hessian) term is dropped in practice, and SAM reports that including second-order terms not only makes the method much less efficient but, surprisingly, degrades robustness performance in their experiments.
>
> Since our method is closely connected to SAM to introduce smoothness into both spaces, we follow the same formulation and also adopt a first-order Taylor approximation to construct perturbations in these two spaces, without adding any stronger assumptions than SAM. Concerning the case where the added perturbation might be relatively large, our parameter studies on perturbation radius (Lines 1397-1402) show that robustness and utility are both best preserved when the radius $\rho$ lies in a moderate range; overly large perturbations hurt utility and are therefore not used in our final setting. In addition, we quantify the Taylor approximation error at each SAM step as the absolute difference between the true loss at the perturbed weights, $L(w+\epsilon)$, and its first-order Taylor approximation, $L(w) + \nabla L(w)^\top \epsilon$. As shown in the **[Figure. R1](https://ibb.co/673CVHxk)**, the resulting relative error ranges from approximately 0.0035\% to 3.39\%, with an average of only about 0.53\%, indicating that the first-order approximation is very tight in practice.

---

> ### Author Response · Authors · 2025-11-24
>
> **W8: Attacker's knowledge**
> Thank you for raising the question regarding attacks. We respectfully clarify that probe-guided forgetting does not receive any information on attackers. As we state in the experimental setup (Line 333-334, Line 1557-1565 and Line 1622-1637), the probe is trained on a held-out split of forget/retain samples from the original dataset, rather than on any prompts or outputs produced by jailbreak attack methods we used in our experiments. The only adversarial component during probe training is a standard FGSM perturbation applied in the representation space, which introduces local smoothness into representation space. Importantly, the whole probe forgetting procedure is not told that there will be a jailbreak attacker, nor does it see any examples from AutoDAN, prefill-based attacks, multi-turn adversarial dialogues or relearn attacks.
>
> **W9: Attacks imbalance**
> We thank the reviewer for pointing out the phenomenon of compromises across different attacks. Building on prior work that interprets attacks as decision margin [3, 13], PRISM does not optimize for a single attack algorithm; instead, it enlarges this margin via smoothness in both parameter and representation space.
>
> Unlike setups where the model is adversarially trained against a specific attack and may overfit to that threat model [14], PRISM is explicitly designed to be attack-agnostic. PRISM is designed to be attack-agnostic rather than tailored to any specific attack. Empirically, we test whether this introduces new weaknesses by evaluating PRISM under four different attacks (one relearning with various sizes and three jailbreak attacks, covering both white-box and black-box settings) (Line 436-445) and by monitoring multiple metrics (unlearning effectiveness, utility, and privacy leakage). Across all these attacks and metrics, PRISM consistently improves robustness without introducing substantial trade-offs. Our ablation study (Line 486-492) shows that removing either smoothness component reduces both robustness and utility, indicating that these components are essential rather than simply trading robustness to one attack for weaknesses elsewhere. While no defense can rule out all possible future attacks, we agree that incorporating an even broader range of attacks is an important direction for future work.

---

> ### Author Response · Authors · 2025-11-24
>
> ### Experiment:
>
> **W10: Response to two-stage training & continuous unlearning**
> We appreciate the reviewer’s question. In our view, the two-stage structure of PRISM (probe+unlearning) is a part of the algorithmic design rather than an unfair advantage. Different unlearning methods adopt different optimization schemes: some are formulated as a single-stage objective, while others also have multi-stage. For example, one of our baselines, DOOR [4], also uses a two-phase procedure (SFT+NPO), and SAM+NPO [3] combines sharpness-aware minimization with NPO. In that sense, PRISM is not unique in using multi-stage optimization; it follows the same principle of designing a training pipeline that matches our motivation.
> Regarding continual unlearning, PRISM can be integrated into an incremental deletion pipeline. Our probe is trained on a held-out set and regularized with FGSM perturbations, so it is robust to small changes in the retain/forget split. Through this, one can reuse the existing probe and simply run the PRISM unlearning stage from the current checkpoint when new requests arrive. If unlearning requests lead to a large distribution shift in the forget set (e.g., new domains or a substantial amount of data), the probe should be retrained, which is cheap in practice since it is a small classifier on top of frozen representations.
>
> **W11, W13, W15, Q2: Baselines, Relearning Samples and WMDP setups**
> We thank the reviewer for highlighting the importance of comparing against strong defense baselines. Our current evaluation already includes several recent and competitive defenses. In particular, we adopt DOOR [4], which specifically targets robustness against jailbreak attacks, and SAM+NPO [3], focusing on robustness under relearning attacks. Across WMDP and MUSE, PRISM consistently achieves higher unlearning scores and stronger robustness to both jailbreak and relearning attacks. To further assess PRISM’s effectiveness, we additionally include RMU [7] and the defense method RMU-LAT [12] in our experiments. In addition, we extend our relearning evaluation by varying the size of the available forget subset. We sample 400 forget samples on MUSE-Books, and then, using different seeds, we sample 200 examples (Relearn50%) and 100 examples (Relearn25%) to run relearning experiments in the rebuttal (Line 466-471, and Line 1540-1547). As shown in the **[Table. R2](https://ibb.co/qY2YLkgf)** and **[Table. R3](https://ibb.co/vCLTjTsX)**, PRISM outperforms RMU and RMU-LAT on the original unlearning setup, Relearn25% and Relearn50% with different steps. After unlearning, RMU shows utility collapse, which can be caused by its random perturbation set to hidden states. Both RMU and RMU-LAT are prone to relearn attacks; their unlearning effectiveness is completely reversed after only 50 steps of relearning attack.
>
> For the original relearning attacks in the initial submission, we used a fixed random seed to sample 400 forget examples as the attack set. For the new Relearn25% and Relearn50% settings, we use different random seeds and sample 100 and 200 forget examples, respectively, from the same pool. This ensures that not only the sample sizes differ, but also the actual contents used in the relearning attacks are different across settings. For the WMDP setups, as discussed in Appendix G.3 (Line 1620-1637), we consider both continuous-text and conversational scenarios. For continuous text (e.g., copyright infrigement where the full text is available), we can use the undesired content directly. For conversational dialogue, however, many undesired contents (such as all harmful biology-related behaviors) cannot be exhaustively listed. To model this, we start from the original WMDP-bio corpus and use GPT-4o to paraphrase it into a QA-style benchmark. To include behavior unlearning, we further include bias-related and refusal-related instances as the forget set, and treat the safe conversations after unlearning as retain data. We also add the cleaned Alpaca dataset as a supplement to the retain set.

---

> ### Author Response · Authors · 2025-11-24
>
> **W12: Response to relatively high overhead**
> We agree that PRISM incurs additional training costs due to the two-stage design. However, this cost comes with substantial gains in both unlearning quality and robustness. First, as shown in Lines 329-335, PRISM significantly improves the unlearning performance compared to NPO, with a ~20% improvement on MUSE-Books and a ~52% improvement on MUSE-News. Second, as shown in Lines 381-388, under 50-step relearning attacks, PRISM reduces verbatim memorization by ~77%, while improving utility by ~43% compared to NPO. Under jailbreak attacks, PRISM outperforms the attack success rate of multi-turn attack by ~10% and prefilling attack by ~8%, as shown in Lines 394-402. Importantly, PRISM has almost the same per-step training time as our strongest main baseline SAM+NPO (11.055s vs. 11.223s), while still achieving better unlearning effectiveness and robustness than SAM+NPO. We believe that, especially in safety-critical settings, these robustness and forgetting–utility balance gains justify the observed training-time overhead. We also included a runtime ablation study **[Table. R4](https://ibb.co/3yHqR7yH)** (Line 1471-1486). All measurements are conducted under the same setup (4 NVIDIA L40s GPUs on a single node, batch size 4, input length 2048 tokens).
>
> **W16, Q1: Response to probe layer selection**:
> We thank the reviewer for the question regarding probe layer choice. As shown in Appendix F.8 (Lines 1418–1431) and **[Table. R5](https://ibb.co/BHHMr0w1)**, we place the probe at early, middle, and late layers on MUSE-Books and report unlearning effectiveness and utility, both with and without relearning attacks. The results indicate that PRISM is not overly sensitive to the exact probe layer: across layers 1–32, the main trends in memorization and utility stay within a reasonable range, and even suboptimal layer choices still outperform SAM+NPO. We do not claim a fully principled procedure for finding the optimal layer; however, both our experiments and prior analyses of transformer representations [15,16] suggest that back layers, which encode richer semantic information, tend to offer a better robustness–utility trade-off, so in practice we recommend selecting the probe from back layers.
>
> **W17: Response to scaling up**
> The largest model supported in our lab environment is an 8B model, so we run unlearning experiments on **Mistral-8B-Instruct-2410**. On this model, as well as three other backbones, we evaluate our dual-space smoothness framework on two datasets, three jailbreak attacks, and multiple steps and sizes of relearning attacks. Across all these settings, PRISM consistently outperforms existing baselines in terms of robustness and forgetting–utility trade-off, while adding only moderate training-time overhead. PRISM is built on smoothness regularization using first-order gradients together with a probe, which is architecture-agnostic. Unfortunately, scaling up to 70B+ models or Mixture-of-Experts architectures is beyond the capacity of our current infrastructure, so we cannot provide direct empirical results in that regime. We view a systematic study of PRISM at the 70B+ scale and on MoE architectures as important future work when sufficient computing resources become available.

---

> ### Author Response · Authors · 2025-11-24
>
> [1] Zhang, Ruiqi, et al. "Negative preference optimization: From catastrophic collapse to effective unlearning." First Conference on Language Modeling.
>
> [2] Shi, Weijia, et al. "Muse: Machine unlearning six-way evaluation for language models." 2025 International Conference on Learning Representations.
>
> [3] Fan, Chongyu, et al. "Towards LLM Unlearning Resilient to Relearning Attacks: A Sharpness-Aware Minimization Perspective and Beyond." Forty-second International Conference on Machine Learning.
>
> [4] Zhao, Xuandong, et al. "Improving llm safety alignment with dual-objective optimization." Forty-second International Conference on Machine Learning.
>
> [5] Foret, Pierre, et al. "Sharpness-aware minimization for efficiently improving generalization." 2021 International Conference on Learning Representations.
>
> [6] He, Zhezhi, et al. "Parametric noise injection: Trainable randomness to improve deep neural network robustness against adversarial attack." Proceedings of the IEEE/CVF conference on computer vision and pattern recognition. 2019.
>
> [7] Li, Nathaniel, et al. "The WMDP Benchmark: Measuring and Reducing Malicious Use with Unlearning." International Conference on Machine Learning. PMLR, 2024.
>
> [8] Shamsian, Aviv, et al. "Go Beyond Your Means: Unlearning with Per-Sample Gradient Orthogonalization." arXiv preprint arXiv:2503.02312 (2025).
>
> [9] Pan, Zibin, et al. "Federated unlearning with gradient descent and conflict mitigation." Proceedings of the AAAI Conference on Artificial Intelligence. Vol. 39. No. 19. 2025.
>
> [10] Hoang, Tuan, et al. "Learn to unlearn for deep neural networks: Minimizing unlearning interference with gradient projection." Proceedings of the IEEE/CVF Winter Conference on Applications of Computer Vision. 2024.
>
> [11] Wang, Yue, et al. "GRU: Mitigating the Trade-off between Unlearning and Retention for LLMs." arXiv preprint arXiv:2503.09117 (2025).
>
> [12] Sheshadri, Abhay, et al. "Latent adversarial training improves robustness to persistent harmful behaviors in llms." arXiv preprint arXiv:2407.15549 (2024).
>
> [13] Lin, Yuping, et al. "Towards understanding jailbreak attacks in llms: A representation space analysis." arXiv preprint arXiv:2406.10794 (2024).
>
> [14] Tramer, Florian, and Dan Boneh. "Adversarial training and robustness for multiple perturbations." Advances in neural information processing systems 32 (2019).
>
> [15] Vulić, Ivan, et al. "Probing pretrained language models for lexical semantics." Proceedings of the 2020 Conference on Empirical Methods in Natural Language Processing.
>
> [16] Rogers, Anna, Olga Kovaleva, and Anna Rumshisky. "A primer in BERTology: What we know about how BERT works." Transactions of the association for computational linguistics 8 (2020): 842-866.

---

### Official Review · Reviewer_6fND · 2025-10-31

**Soundness:** 2
**Presentation:** 2
**Contribution:** 2
**Rating:** 4
**Confidence:** 2

**Summary:**

This paper proposes PRISM, a machine unlearning framework that enforces smoothness in the representation space and parameter space. In the representation space, they adversarially train a robust classifier that separates the representation of the forget set and the retain set. Then, they consider the minimax of a perturbed loss that combines the NPO loss and a guidance term from the trained robust classifier. When finetuning on this loss, they project the gradient to the orthogonal space of the gradient of the retain set, to protect utility. They also apply Sharpness Aware minimization here to reach smooth regions. On two models and three datasets, they show their framework achieves better Unlearn Score (in terms of verbatim memorization), while preserving utility and improving robustness to relearning (for up to 100 steps) and multiple jailbreak attacks.

**Strengths:**

- Using adversarial robustness for unlearning in both representation space (for defensing jailbreak attacks) and the parameter space (for relearning attacks) is an interesting idea.
- Their empirical evaluations corroborate the claimed performance (better unlearning utility tradeoff, and robustness against these attacks) of the proposed method.

**Weaknesses:**

- Several claims are not directly validated.  For example, is it possible to quantify margins for small datasets to directly show that PRISM widens the margin in the representation space? There is also limited discussion on the probe capacity, pooling, and sensitivity to the adversarial radius. This makes the robustness claim a bit unclear.

- Relearning robustness is varied by steps only. The evaluation does not sweep the size of the available forget subset.

Minor clarity issue: $\mathcal{L}_{\text{gen}} $ is not explicitly defined in the main text

**Questions:**

PRISM’s representation-space component hinges on a robust probe separating forget vs retain features at some layer. How does the method behave when these distributions are not separable at any layer (e.g., PII removal where forget and retain are near-identical except for personal tokens)? In this setting, can the probe meaningfully guide updates, or does it worsens the unlearning score? Does the orthogonal projection simply stall progress when gradients are highly colinear?

---

> ### Author Response · Authors · 2025-11-24
>
> We thank Reviewer 6fND for the helpful and constructive reviews. Below, we address each key point raised in the comments. We also add additional experiments' results through **anonymous** links for your reference.
>
> **W1: Response to quantifying margins**
> We sincerely thank the reviewer for these helpful suggestions. In the revised version, we add more details on the probe setup (capacity, pooling, and adversarial radius) in (Line 1185-1195). In our original submission's Appendix F.7 (Line 1404-1413), we included a parameter study of adversarial radius in MUSE-News w/ and w/o relearning attacks. From the study, we found that a small radius already gives strong forgetting and good clean utility but weak protection against relearning.  Large radius improves robustness to relearning but hurts utility and stability. Thus, in our training, we chose a moderate adversarial radius to balance among the critic metrics.
>
> In the rebuttal, we add an empirical study that quantifies the margins in the representation space (Line 472-479 and 1512-1529). Concretely, for the unlearned model, we extract layer-L representations of forget examples and feed them into a frozen adversarial probe, whose last linear layer defines a decision hyperplane between harmless (class 0) and harmful (class 1) outputs. We compute the margin as the distance from each representation to this hyperplane, using the logit difference between harmless and harmful classes divided by the norm of the weight vector. As shown in **[Table R.1](https://ibb.co/zT0XdX3X)**, compared to the original model, PRISM yields a substantially wider margin (median margin +24.9% and 10%-quantile margin ≈4.1×) and also consistently outperforms SAM+NPO on both metrics. This shows that PRISM enlarges the representation-space margin and lifts the low-margin tail, which clarifies how our representation-space smoothness contributes to jailbreak robustness, as discussed in [1].
>
> **W2&W3: Response to various relearn sizes**
> We thank the reviewer for the valuable suggestion. In the revised version, we have added experiments that vary the size of the available forget subset in the relearning attack to provide a more comprehensive evaluation (Line 465-471 and Line 1541-1547). Initially, we used 400 samples for the Books dataset in the relearning attack. In the additional experiments, we also evaluate using 100 samples and 200 samples to compare the effects of different subset sizes. Additionally, we include two new baselines, RMU [2] and RMU-LAT [3], to further enhance the comparison and demonstrate the effectiveness of our approach across different settings. As shown in (**[Relearn25%](https://ibb.co/qY2YLkgf)**) and (**[Relearn50%](https://ibb.co/vCLTjTsX)**), RMU and RMU-LAT are almost completely reversed by the relearning attacks, with “No VerbMem/No KnowMem” increasing sharply after 50/100 steps, while PRISM consistently keeps memorization much lower and utility high, outperforming all baselines in robustness under both Relearn25% and Relearn50% settings. Regarding W3, here we define $L\_gen$ that
> downweights preferences on the forget set with the reference model, following [4]. We also updated this in Line 292.

---

> ### Author Response · Authors · 2025-11-24
>
> **Q1, Q2 & Q3: Response to PII removal and colinear gradients**
> We thank the reviewer for this insightful question. We believe that even in PII-style settings, forget and retain features are not completely non-separable. For example, [5] replaces only person, organization, or country names while keeping the rest of the sentence fixed and reports that 25–35% of predictions change, which shows that small changes in PII tokens cause clear shifts in hidden representations and outputs. [6] shows that models leak specific PII strings after PII scrubbing, and [7] shows that models can relearn unique email addresses when trained later on data with overlapping n-grams. These results indicate that PII tokens create structured variation in the representation space rather than being indistinguishable. This is consistent with our layer study at Appendix F.8 (Line 1419-1431), where probes in back layers, which encode more semantic information, yield a better unlearning performance. These suggest that the probe can still provide meaningful guidance for smoothness and unlearning in such scenarios. In truly extreme cases where forget and retain distributions are indistinguishable at all layers, the probe can be viewed as introducing almost random perturbations. Our ablation study (Line 488-492), where representation-space smoothness is removed, shows that PRISM still achieves effective unlearning, which suggests that the method degrades gracefully to its parameter-space component. More importantly, we see a more systematic study of such PII-focused settings as an important direction for future work in machine unlearning.
>
> Regarding the concern that orthogonal projection might stall progress when gradients are highly colinear, we respectfully note that the forget gradient is updated using a SAM-style adversarial perturbation, which introduces worst-case perturbations to the gradient (Line 298-304). After this perturbation, the forget gradient is sufficiently different from the retain gradient. Even in an extreme mini-batch where the two gradients are nearly collinear, this just reduces the step size along the conflicting direction, rather than turning the update to zero. The remaining orthogonal component of the forget gradient, the SAM-based smoothness update, and the gradients from subsequent mini-batches with different directions continue to update the parameters.
>
> [1] Lin, Yuping, et al. "Towards understanding jailbreak attacks in llms: A representation space analysis." arXiv preprint arXiv:2406.10794 (2024).
>
> [2] Li, Nathaniel, et al. "The WMDP Benchmark: Measuring and Reducing Malicious Use with Unlearning." International Conference on Machine Learning. PMLR, 2024.
>
> [3] Sheshadri, Abhay, et al. "Latent adversarial training improves robustness to persistent harmful behaviors in llms." arXiv preprint arXiv:2407.15549 (2024).
>
> [4] Zhang, Ruiqi, et al. "Negative preference optimization: From catastrophic collapse to effective unlearning." arXiv preprint arXiv:2404.05868 (2024).
>
> [5] Balasubramanian, Sriram, et al. "What's in a Name? Are BERT Named Entity Representations just as Good for any other Name?." arXiv preprint arXiv:2007.06897 (2020).
>
> [6] Lukas, Nils, et al. "Analyzing leakage of personally identifiable information in language models." 2023 IEEE Symposium on Security and Privacy (SP). IEEE, 2023.
>
> [7] Borkar, Jaydeep, et al. "Privacy Ripple Effects from Adding or Removing Personal Information in Language Model Training." arXiv preprint arXiv:2502.15680 (2025).

---

### Official Review · Reviewer_hH6e · 2025-10-31

**Soundness:** 3
**Presentation:** 4
**Contribution:** 3
**Rating:** 8
**Confidence:** 3

**Summary:**

This paper introduces PRISM (Probe-guided Iterative Smoothness Minimization) — a novel dual-space unlearning framework for large language models (LLMs). The key idea is to enforce smoothness in both representation and parameter spaces to achieve robustness against jailbreak and relearning attacks while balancing unlearning effectiveness, utility preservation, and privacy protection.
Experiments on MUSE-Books, MUSE-News, and WMDP-bio show PRISM consistently outperforms baselines (NPO, SAM+NPO, DOOR, Task Vector) in both unlearning performance and robustness under adversarial attacks.

**Strengths:**

Unified framework: Bridges representation-space and parameter-space robustness under a single smoothness-driven formulation.

Balanced performance: Demonstrates superior trade-offs among unlearning effectiveness (UE), post-unlearning performance (PP), and privacy leakage.

Strong empirical validation: Outperforms all baselines across MUSE and WMDP tasks, with consistent robustness to both jailbreak and relearning attacks.

Comprehensive evaluation: Includes ablations (Table 4), parameter studies, and diverse attack types.

Reproducibility: Code and dataset details are clearly provided.

**Weaknesses:**

The formal connection between probe-based representation smoothness and overall model generalization could be elaborated more rigorously.

No ablation on the computational overhead of dual-space optimization.

**Questions:**

How sensitive is PRISM to the probe layer selection? Could higher transformer layers generalize better for representation-space smoothness?

Can PRISM be extended to behavioral unlearning (e.g., refusal tuning or bias mitigation)?

What is the empirical trade-off between unlearning speed and robustness when scaling to 70B+ models?

Does the probe retain knowledge of harmful content after training, potentially introducing privacy concerns?

Could PRISM integrate into a continual unlearning pipeline (e.g., for incremental data deletions)?

---

> ### Author Response · Authors · 2025-11-24
>
> We thank Reviewer hH6e for the thorough review and the encouraging comments on our contributions and presentation. We also greatly appreciate the constructive feedback. Below, we address each key point raised in the comments. We also add additional experiments' results through **anonymous** links for your reference.
>
> **W1: Representation smoothness formal analysis**
> Thank you for raising this question. We include a formal analysis in the appendix of the original submission (Line 1008-1025). In particular, we show that, under an $\ell_\infty$ perturbation budget, the inner adversarial maximization admits a closed-form FGSM solution. Under a standard $\beta-Lipschitz$ smoothness assumption on the loss, we further prove that the gap between the true worst-case loss and its first-order approximation is bounded by $O(d\varepsilon^2)$, which gives an explicit upper bound on the adversarial loss increase. Enforcing probe-based representation smoothness tightens this bound and improves robustness to jailbreak attacks. It also stabilizes the model’s predictions, which in turn leads to better overall generalization in our unlearning setting.
>
> **W2&Q3: Runtime ablation & Scale up**
> We thank the reviewer for pointing out the need for a clearer runtime analysis. We have added a runtime ablation experiment **([Table. R1](https://ibb.co/3yHqR7yH))** that measures the extra cost introduced by each PRISM component (Line 1480-1486). All methods are evaluated under the same setup (4 NVIDIA L40S GPUs on a single node, batch size 4, input length 2048 tokens). From Table. R1, we see that parameter smoothness (PS) causes the largest increase in runtime, adding 35.45% to the time per step. This is because PS requires searching for a worst-case perturbation when updating the parameters, which naturally takes more computation.
>
> The largest model supported in our lab environment is **Mistral-8B-Instruct-2410**. On this model, as well as three other backbones, we evaluate our dual-space smoothness framework on two datasets, three jailbreak attacks, and multiple steps and sizes of relearning attacks. Across all these settings, PRISM consistently outperforms existing baselines in terms of robustness and trade-off, while adding only moderate training-time overhead. PRISM is built on smoothness regularization using first-order gradients together with a probe, which is architecture-agnostic. Unfortunately, scaling the full experimental suite to 70B+ models is beyond the capacity of our current infrastructure, so we cannot provide direct empirical measurements in that regime. We view a systematic study of PRISM at the 70B+ scale as important future work when sufficient computing resources become available.
>
> **Q1: Response to probe layer selection**:
> We sincerely thank the reviewer for this insightful question and for highlighting the potential importance of the probe layer. We have included a detailed layer selection study in our original submission in Appendix F.8 (Lines 1415–1431), as shown in **[Table. R2](https://ibb.co/BHHMr0w1)**. We place the probe at different layers spanning the front, middle, and back segments on MUSE-Books, and report metrics on unlearning effectiveness and utility w/ and w/o relearning attacks. As we can see, the results show that PRISM is not overly sensitive to the exact probe layer: across layers 1–32, the main trends of memorization and utility vary within , and even less optimal layers still outperform SAM+NPO.
>
> We also appreciate the reviewer’s insightful intuition that higher layers generally offer a better robustness–utility trade-off. Our layer study supports this view. Probes placed in the back layers  (e.g., layers 23/28/32) tend to achieve lower VerbMem and KnowMem after 100-step relearning, while maintaining comparable or slightly better utility than earlier layers. This observation is consistent with prior analyses of transformer representations [1][2], which show that higher layers aggregate more contextual and semantic information. Thus, higher layers are suitable for enforcing representation-space smoothness in PRISM.

---

> > ### Comment · Reviewer_hH6e · 2025-11-27
> >
> > W2: The added runtime table helps to explain the runtime overhead.
> >
> > Q3:  It is tricky to adapt the PRISM to real system for the concern of too expensive or too heavy. A discussion about memory scaling, distributed training stability, or how expensive it becomes as model size grows will enlarge the contribution.
> >
> > Q1: The response is fair for probe layer selection and shows the method isn’t super sensitive.

---

> ### Author Response · Authors · 2025-11-24
>
> **Q2&Q5: Response to unlearning setups**
> We thank the reviewer for highlighting these two unlearning scenarios. Regarding behavioral unlearning, in our experiments, we already adopt conversational dialogue settings, where the unlearning set includes harmful or biased behaviours (e.g., unsafe or policy-violating QA pairs from safety and bias benchmarks) as the forget set and safe behaviours as the retain set (Line 1627-1637). We then train the probe and our dual-space smoothness objective on this behavioural partition. Thus, our conversational experiments already treat PRISM as a form of behavioural unlearning, where the goal is to remove undesired behaviours while preserving safe dialogue capabilities.
>
> Regarding continual unlearning, PRISM can be integrated into a pipeline with incremental deletions. Given a new unlearning request, one can (i) update or re-train the probe on the current retain/forget split and (ii) run the unlearning stage from the current checkpoint using PRISM. The probe is lightweight and only used during training, so training can be repeated as new deletion requests arrive without changing inference-time cost. Our experiments, however, focus on the single-shot unlearning setting commonly used in the community, and we do not claim to fully address continual unlearning. We view extending PRISM to large-scale continual unlearning as important future work.
>
> **Q4: Response to probe's privacy concerns**:
> We thank the reviewer for raising this important privacy question. In our setting, the probe is a small MLP defined on top of frozen representations, and is only used as an auxiliary module during training to construct perturbations. It is not a generative model and does not operate in the token space, so it does not store or reproduce verbatim harmful content. After the unlearning process, the probe is not used at inference time and can be safely discarded. Only the updated base model is deployed. Thus, the probe does not introduce additional privacy risks beyond those of standard unlearning training.
>
>
> [1] Vulić, Ivan, et al. "Probing pretrained language models for lexical semantics." Proceedings of the 2020 Conference on Empirical Methods in Natural Language Processing.
>
> [2] Rogers, Anna, Olga Kovaleva, and Anna Rumshisky. "A primer in BERTology: What we know about how BERT works." Transactions of the association for computational linguistics 8 (2020): 842-866.

---

> > ### Comment · Reviewer_hH6e · 2025-11-27
> >
> > Q2 & Q5: The response from author "How PRISM could be used for continual unlearning" makes sense—the probe can be retrained and the unlearning run again incrementally.
> >
> > Q4: This question is addressed properly.
> >
> > Thanks for the response. I will keep my positive rating.

---

### Official Review · Reviewer_3Qb5 · 2025-11-01

**Soundness:** 2
**Presentation:** 2
**Contribution:** 2
**Rating:** 4
**Confidence:** 4

**Summary:**

The paper proposes PRISM, a unified framework to enhance the robustness and balance of large language model (LLM) unlearning. Existing methods often over-optimize one objective—like forgetting effectiveness or utility preservation—while remaining vulnerable to relearning and jailbreak attacks. PRISM tackles these issues by enforcing dual-space smoothness across both the representation and parameter spaces. In the representation space, a robustly trained probe defends against jailbreak manipulations by promoting feature-level smoothness. In the parameter space, Sharpness-Aware Minimization (SAM) and gradient decoupling between retain and forget objectives ensure stability and prevent the model from relearning forgotten knowledge. Extensive experiments on WMDP and MUSE benchmarks show that PRISM outperforms prior unlearning baselines (e.g., NPO, GA, DOOR, SAM+NPO) with improved robustness, better forgetting–utility balance, and higher resistance to adversarial attacks.

**Strengths:**

1.	The paper introduces a novel dual-space smoothness framework (PRISM) that jointly regularizes representation and parameter spaces, offering a fresh and well-motivated perspective on robust LLM unlearning.
2.	The min–max formulation and incorporation of Sharpness-Aware Minimization (SAM) are conceptually sound and effectively connected to the goal of preventing both jailbreak and relearning attacks.
3.	The paper conducts comprehensive experiments on multiple benchmarks (WMDP, MUSE) and clearly demonstrates improved robustness and better trade-offs between unlearning, utility, and stability compared with prior methods.

**Weaknesses:**

1.	The method introduces considerable architectural and computational complexity (e.g., robust probe, dual-space optimization, gradient orthogonalization) without providing a clear runtime or memory analysis to justify the added cost.
2.	The comparative baselines are somewhat limited and do not include more recent or parameter-efficient unlearning approaches, making it difficult to fully assess PRISM’s relative advantage.
3.	Some theoretical explanations lack formal grounding, particularly the connection between representation-space smoothness and jailbreak robustness, which is presented intuitively but without rigorous analysis or ablation evidence.

**Questions:**

1.	How sensitive is PRISM to the choice of layer used for the representation-space probe? Would using deeper or earlier layers change the trade-off between forgetting and utility?
2.	Could the authors clarify the mathematical connection between representation-space smoothness and jailbreak robustness? Is there any empirical ablation isolating the probe’s contribution?
3.	How does PRISM perform under multiple sequential unlearning requests or when the forget set is large? Does smoothness enforcement accumulate interference?
4.	Could the authors provide ablation results under the orthogonal gradient setting when each smoothness component (PS or RS) is removed individually? It would be helpful to see how the model behaves when parameter-space smoothness or representation-space smoothness is independently disabled, to better understand their respective contributions beyond the combined RS + GCD ablation shown in Table 4.

---

> ### Author Response · Authors · 2025-11-24
>
> We thank Reviewer 3Qb5 for the helpful and constructive reviews. Below, we address each key point raised in the comments. We also add additional experiments' results through **anonymous** links for your reference.
>
>
>
> **W1: Response to runtime analysis**
> We thank the reviewer for pointing out the need for a clearer runtime analysis. We have added an ablation runtime experiment (Line1480-1486) **([Table. R1](https://ibb.co/3yHqR7yH))** that explicitly measures the incremental overhead of each PRISM component. All measurements are conducted under the same setup (4 NVIDIA L40s GPUs on a single node, batch size 4, input length 2048 tokens). As we can see, the parameter smoothness (PS) contributes the most to the increase in runtime, adding 35.45% to the time per step. This is due to the complexity of searching for the worst-case perturbation when updating the parameters, which requires more computational effort.
>
> **W2: Comparative baselines**
> We thank the reviewer for this comment. In our experiments, we already include both classical unlearning baselines (GA, NPO, Task Vector) and more recent SOTAs such as DOOR and SAM+NPO. To further strengthen the comparison, in the revised version   (Line 1500-1508) we additionally include RMU [1] and RMU-LAT [2] as baselines and conduct corresponding unlearning experiements **([Table. R2](https://ibb.co/GfvJPzmJ))**. As shown in the table, PRISM significantly outperforms both RMU and RMU-LAT in terms of unlearning effectiveness and utility preservation.
>
> We also further conduct relearning attacks on these two new baselines, as shown in Line 455-465 (**[Table. R4](https://ibb.co/qY2YLkgf)**). The RMU family's unlearning effectiveness are almost completely reversed by the relearning attacks, with “No VerbMem/No KnowMem” increasing sharply after 50/100 steps, while PRISM consistently keeps memorization much lower and utility high, outperforming other baselines in both robustness and utility.
>
> **W3, Q2: Mathematical Connection and Empirical Study**
> This is a very intriguing comment, and we have provided a formal analysis in the appendix of the original submission (Line 1008-1025). There, we show that, under an $\ell_\infty$ perturbation budget, the inner maximization of the loss admits a closed-form FGSM solution. Assuming a $\beta$-Lipschitz gradient, we further prove that the gap between the true worst-case loss and its first-order proxy is bounded by $O(d\varepsilon^2)$. This leads to an explicit upper bound as the adversarial loss increase. Enforcing representation-space smoothness directly tightens this bound and thus improves robustness to jailbreak-style perturbations.
>
> We further emperically measure the geometric margin in the representation space using the frozen adversarial probe (Line 472-479). Specifically, for the unlearned model we extract layer-L representations of forget examples, feed them into the probe, and treat the probe’s last linear layer as defining a decision hyperplane between the harmless and harmful classes. We then compute the margin as the distance from each representation to this hyperplane. As shown in the table **([Table. R3](https://ibb.co/zT0XdX3X))**, compared to the original model, PRISM yields a substantially wider margin, increasing the median margin by 24.9% and the 10%-quantile margin by ≈4.1×, and it also consistently outperforms SAM+NPO on both metrics. This shows that PRISM widens the average representation-space margin and lifts the low-margin tail. Thus, representation-space smoothness that enlarges the margin can improve jailbreak robustness, as discussed in [3].
>
> **Q1: Probe Layer Selection**
> We thank the reviewer for this question regarding the sensitivity of PRISM to the choice of probe layer. In Appendix F.8 of our original submission (Lines 1415–1431) and **[Table. R5](https://ibb.co/BHHMr0w1)**, we provided a detailed layer-selection study on MUSE-Books, where we place the probe at various layers in the early, middle, and late parts of the network and report unlearning effectiveness and utility, both with and without relearning attacks. The results indicate that PRISM is not overly sensitive to the precise probe layer: across layers 1–32, the main trends in memorization and utility remain within a reasonable range, and even suboptimal layer choices still outperform SAM+NPO. Moreover, our study shows that higher layers generally offer a better robustness–utility trade-off. Probes placed in the back layers (e.g., layers 23/28/32) tend to achieve lower VerbMem and KnowMem after 100-step relearning, while maintaining competitive utility. This is consistent with prior analyses of transformer representations [4, 5], which show that higher layers aggregate richer contextual and semantic information, making them suitable for our probe-based regularization.

---

> ### Author Response · Authors · 2025-11-24
>
> **Q3: Scale of Forget Set**
> Thank you for the question. Our experiments mainly focus on the standard setting commonly used in the community, which is a single unlearning request with a fixed forget set (Line 1558-1565). We benchmark on MUSE and WMDP with a single unlearning request and a large forget set (e.g., full Harry Potter books, BBC news, and various harmful WMDP queries). In this regime, PRISM consistently achieves the highest Unlearn Score, indicating that dual-space smoothness does not cause catastrophic interference. Also, once forget examples are pushed into a “safe” region, the gradients from both representation- and parameter-space smoothness naturally decay, so the smoothness signal saturates rather than accumulating interference.
>
> **Q4: Ablation Study**
> We thank the reviewer for the helpful suggestion. In the original version, we already ablated PS, GCD, and GCD&RS (Line 486-493), and in the rebuttal, we additionally report the RS ablation. As shown in **[Table. R6](https://ibb.co/r2QGRbYn)** , removing PS significantly increases VerbMem on $D_f$ under relearning attacks (with the largest rise at 100 steps), and removing RS likewise leads to higher VerbMem and lower utility, showing that both smoothness components are crucial to robustness under the orthogonal gradient setting.
>
>
> [1] Li, Nathaniel, et al. "The WMDP Benchmark: Measuring and Reducing Malicious Use with Unlearning." International Conference on Machine Learning. PMLR, 2024.
>
> [2] Sheshadri, Abhay, et al. "Latent adversarial training improves robustness to persistent harmful behaviors in llms." arXiv preprint arXiv:2407.15549 (2024).
>
> [3] Lin, Yuping, et al. "Towards understanding jailbreak attacks in llms: A representation space analysis." arXiv preprint arXiv:2406.10794 (2024).
>
> [4] Vulić, Ivan, et al. "Probing pretrained language models for lexical semantics." Proceedings of the 2020 Conference on Empirical Methods in Natural Language Processing.
>
> [5] Rogers, Anna, Olga Kovaleva, and Anna Rumshisky. "A primer in BERTology: What we know about how BERT works." Transactions of the association for computational linguistics 8 (2020): 842-866.

---

### Comment · Area_Chair_gwrT · 2025-11-27

Dear Reviewers,

The discussion phase will end soon. The authors have provided responses to your comments. Please check if there are more concerns by adding official comments.

Thanks for your efforts and contributions to ICLR 2026.

Best regards,

Your Area Chair

---

### Author Response · Authors · 2025-12-02
**Key takeaways from our responses to reviewers (1/2)**

Dear Area Chair and Reviewers,

Thank you very much for your efforts in overseeing and managing the review process, particularly during this challenging time. We **have not received any responses** from 3 of our reviewers after submitting the rebuttal and the revision. And only Reviewer hH6e responded to our rebuttal and maintained the positive score. Thus, we would like to briefly summarize our replies and highlight several key takeaways. We also add additional experiments' results through **anonymous** links for your reference. We believe that we have successfully addressed every issue raised during the review process, thereby significantly enhancing the paper.

**Takeaway 1: Additional baselines and various forget set sizes during attacks** (R-3Qb5's W2, R-6fND's W2, and R-KUGw's W11&W13&W15)

Besides the classic baselines, our original submission includes recent and competitive defenses. In particular, we adopt DOOR [1], which targets robustness against jailbreak attacks, and SAM+NPO [2], focusing on robustness under relearning attacks. Initially, we used 400 samples for the Books dataset in the relearning attack. In the rebuttal, we also evaluate using 100 samples and 200 samples to compare the effects of different subset sizes. Additionally, we include two new baselines, RMU [3] and RMU-LAT [4], to further enhance the comparison.

Here we attach the **[Relearn25%](https://ibb.co/qY2YLkgf)** (100 samples) in the above table and **[Relearn50%](https://ibb.co/vCLTjTsX)** (200 samples) in the link. As shown in the table, RMU and RMU-LAT are almost completely reversed by the relearning attacks, while PRISM consistently keeps memorization much lower and utility high, outperforming all baselines in robustness.

**Takeaway 2: Probe Layer Selection** (R-3Qb5's Q1, R-hH6e's Q1 and R-KUGw's W16&Q1)

We have included a layer selection study in our original submission in Appendix F.8 (Lines 1415–1431). We place the probe at layers spanning the front, middle, and back segments on MUSE-Books, and report metrics on unlearning effectiveness and utility w/ and w/o relearning attacks. The results show that PRISM is not overly sensitive to the exact probe layer.

Our layer study shows that higher layers generally offer a better robustness–utility trade-off, which is consistent with prior analyses of transformer representations. Probes placed in the back layers  (e.g., layers 28/32) tend to achieve lower VerbMem and KnowMem after 100-step relearning, while maintaining comparable or slightly better utility than earlier layers.

**Takeaway 3: Continual and behavioral unlearning setups** (R-3Qb5's Q3, R-hH6e's Q2&Q5, R-KUGw's W10&Q2)

PRISM supports continual unlearning. The probe is trained on a held-out set with FGSM regularization; it is robust to small distribution shifts, so one can reuse the probe and rerun the PRISM unlearning stage from the current checkpoint when new requests arrive. If later unlearning requests cause a large shift in the forget distribution, the probe can be cheaply retrained as a small classifier on frozen representations. As also noted by Reviewer hH6e, this design “makes sense—the probe can be retrained and the unlearning run again incrementally.”

We adopt conversational dialogue settings, where the unlearning set includes harmful or biased behaviours (e.g., unsafe pairs from WMDP\_bio) as the forget set and safe conversations as the retain set (Line 1627-1637). We then train the probe and our dual-space smoothness objective on this behavioural partition. Thus, our conversational experiments already treat PRISM as a form of behavioural unlearning, where the goal is to remove undesired behaviours while preserving safe ones.

**Takeaway 4: Ablation study and scale up** (R-3Qb5's W1, R-hH6e's W2&Q3 and R-KUGw's W12&W17)

We have added a runtime ablation experiment that measures the extra cost introduced by each PRISM component (Line 1480-1486), as listed in **[Table. T1](https://ibb.co/3yHqR7yH)**. The parameter smoothness (PS) contributes the most to the increase in runtime. Although PRISM introduces additional training costs due to its two-stage design, this overhead is compensated by substantial gains in both unlearning effectiveness and robustness. For example, compared to NPO, PRISM achieves roughly a 20% improvement in unlearning effectiveness and up to ~77% improvement on certain robustness metrics.

We ablated components in the original submission (Line 486-493), and in the rebuttal we additionally report the Representation Smoothness ablation. As shown in **[Table. T2](https://ibb.co/r2QGRbYn)**, removing RS causes vulnerability to attacks, showing that both smoothness components are crucial to robustness.

In our experiments on the four largest backbones currently supported in our lab, PRISM consistently achieves strong robustness and forgetting–utility balance, though scaling to 70B+ and MoE models remains future work due to current compute limits.

---

### Author Response · Authors · 2025-12-02
**Key takeaways from our responses to reviewers (2/2)**

**Takeaway 5: Representation Margin** (R-3Qb5 Q2, R-hH6e W1, R-6fND W1, R-KUGw W7)

In our original submission, we show that under an $\ell_\infty$ budget the inner maximization admits a closed-form FGSM solution with adversarial loss increase bounded by $O(d\varepsilon^2)$, and that enforcing probe-based representation smoothness tightens this bound, improving robustness and generalization.

In the rebuttal, we add an empirical study that measures representation-space margins as the distance from layer-L representations of forget examples to a probe’s decision hyperplane. As shown in **[Table. T3](https://ibb.co/zT0XdX3X)**, PRISM substantially enlarges the median margin (+24.9%) and lifts the low-margin tail (≈4.1× at the 10% quantile) over the original model and SAM+NPO.

**Takeaway 6: Probe privacy and inseparable distribution** (R-hH6e's Q4, R-6fND's Q1-Q3)

Reviewers raised two concerns about the probe: potential privacy risks and its usefulness when forget/retain distributions are hard to distinguish. In our framework, the probe is a small non-generative MLP trained on frozen, so it does not store or reproduce harmful text. It can be discarded after unlearning, and therefore does not introduce additional privacy risks. Reviewer hH6e considered this question **properly addressed**. In PII-style settings, prior work and our layer-wise probe analysis show that PII tokens lead to structured changes in hidden representations, so the probe still offers meaningful smoothness and unlearning guidance. In the extreme case where forget and retain are indistinguishable, the probe effectively becomes near-random noise, and our ablation without representation-space smoothness also shows that PRISM still unlearns effectively via its parameter-space component.

**Conclusion**

The reviewers collectively recognize the significance and novelty of our work, describing it as “a novel dual-space smoothness framework, offering a fresh and well-motivated perspective on robust LLM unlearning” (3Qb5) and noting that enforcing smoothness “in both representation space and the parameter space is an interesting idea” (6fND). The consensus is that our framework of smoothness minimization in both spaces “are conceptually sound and effectively connected to the goal of preventing both jailbreak and relearning attacks” (3Qb5) and “bridges representation-space and parameter-space robustness under a single smoothness-driven formulation” (hH6e), and that the “comprehensive experiments on multiple benchmarks” (3Qb5) “corroborate the claimed performance of the proposed method.” (6fND)

With the additional extensive experiments and provided clarifications during the rebuttal, we believe the remaining weaknesses regarding baselines, relearn set sizes, runtime study, and empirical study of our theoretical contributions have been fully resolved. Even in the absence of further discussion with the reviewers, we sincerely appreciate the Area Chairs for their extra time, effort, and thoughtful handling of our submission. **Thank you once again for your careful consideration.**

---

> ### Author Response · Authors · 2025-12-02
> **References**
>
> [1] Zhao, Xuandong, et al. "Improving llm safety alignment with dual-objective optimization." Forty-second International Conference on Machine Learning.
>
> [2] Fan, Chongyu, et al. "Towards LLM Unlearning Resilient to Relearning Attacks: A Sharpness-Aware Minimization Perspective and Beyond." Forty-second International Conference on Machine Learning.
>
> [3] Li, Nathaniel, et al. "The WMDP Benchmark: Measuring and Reducing Malicious Use with Unlearning." International Conference on Machine Learning. PMLR, 2024.
>
> [4] Sheshadri, Abhay, et al. "Latent adversarial training improves robustness to persistent harmful behaviors in llms." arXiv preprint arXiv:2407.15549 (2024).

---

### Meta-Review · Area_Chair_8jmG · 2026-01-07

**Summary:**

space smoothness framework for robust LLM unlearning, aiming to improve resistance to both jailbreak and relearning attacks while preserving utility. During the rebuttal, the authors addressed the majority of reviewer concerns by adding new baselines, expanding robustness and relearning evaluations, clarifying notation and threat models, and providing additional empirical analyses. Most of the remaining issues are either minor or could be addressed during the camera-ready stage.

The main outstanding concern, primarily raised by Reviewer KUGw (R4), relates to the conceptual novelty of the proposed “dual-space” formulation. While parameter-space and representation-space smoothness are each well-motivated and empirically effective, the paper does not fully establish that jointly enforcing both yields synergistic benefits beyond combining two independently motivated objectives. In particular, given that jailbreak and relearning attacks are defined under different threat models, it remains unclear whether PRISM demonstrates a principled dual-space synergy or primarily a successful combination of multiple robustness techniques. This raises a substantive concern about “dual-space” framing.

Nevertheless, the paper presents a coherent and well-engineered algorithm that successfully integrates multiple robustness mechanisms without internal contradictions, and demonstrates strong empirical performance across several benchmarks and attack settings. While the dual-space novelty could be further highlighted, the overall contribution is solid enough to merit acceptance at the poster level.

Finally, regardless of the strengths aforementioned, the presentation of the paper remains somewhat difficult to follow. The combination of the two distinct sub-problems and earlier notation issues may have likely contributed to reviewers’ mixed assessments of presentation quality. If accepted, the AC strongly encourages the authors to substantially improve clarity and organization in the camera-ready version.

**Reviewer Concerns:**

Below we summarize the key reviewer concerns and indicate which issues were addressed by the rebuttal, which were partially addressed, and which remain outstanding.

### Concerns largely addressed by the rebuttal

1. **Missing or insufficient baselines (R1, R4).**
Reviewers noted that several important recent unlearning baselines, including robustness-oriented methods, were missing. The rebuttal added additional competitive baselines such as RMU and RMU-LAT, substantially strengthening the empirical comparison.

2. **Lack of quantitative evidence for representation-space margin improvement (R3).**
The original submission did not quantitatively demonstrate how much representation-space margins were widened, especially on small datasets. The authors added explicit margin measurements and tail analysis, clarifying the empirical basis of their robustness claims.

3. **Insufficient discussion of probe design choices such as probe capacity, pooling strategy, and adversarial radius (R3).**
The rebuttal significantly expanded discussion and analysis of these factors. While not exhaustive, the added explanations and parameter studies are sufficient for the scope of the paper.

4. **Relearning robustness evaluated only by attack steps, not forget-set size (R3, R4).**
Reviewers pointed out that relearning attacks were varied only by the number of steps and that forget-set sizes were unclear. The rebuttal added experiments sweeping different forget subset sizes (e.g., 25% and 50%) with multiple seeds.

5. **Notation issues and missing definitions (R3, R4).**
Several symbols (e.g., g, h, D_a, L_"gen" ) were previously undefined or inconsistently used. The authors clarified notation and added missing definitions, resolving the main sources of confusion.

6. **Unclear threat model (R4).**
The rebuttal clarified the threat models assumed for jailbreak attacks (black-box query access) and relearning attacks (white-box access to model weights), making the evaluation setting more explicit.

7. **Validity of first-order Taylor approximation under perturbations (R4).**
Reviewers questioned whether first-order approximations remain valid for non-negligible perturbations. The authors justified this choice by drawing parallels to SAM and provided empirical measurements showing that approximation errors are small in practice.

8. **Concern that probe-guided forgetting assumes knowledge of the attacker (R4).**
The rebuttal clarified that the probe is trained on held-out forget/retain data and does not use jailbreak prompts or attacker-specific information, making the approach attack-agnostic.

9. **Sensitivity to probe layer selection (R1, R2, R4).**
Additional experiments showed that PRISM is relatively robust to probe layer choice, with later layers typically offering a better robustness–utility trade-off.

10. **Applicability to behavioral unlearning (R2).**
The authors clarified that their conversational unlearning experiments already cover behavioral unlearning scenarios, addressing this concern.

11. **Probe privacy concerns (R2).**
The probe is non-generative, used only during training, and discarded afterward. While subtle indirect effects cannot be entirely ruled out, the reviewer’s privacy concern was adequately addressed.

12. **Behavior under near-inseparable forget/retain distributions (e.g., PII removal) (R3).**
The rebuttal provided a plausible explanation that PRISM degrades gracefully to its parameter-space component when representation-space guidance becomes weak.

13. **Potential stalling due to near-colinear gradients in orthogonal projection (R3).**
The authors argued that SAM-style perturbations and batch variability prevent complete stalling, which is a reasonable explanation at this level.

### Concerns judged not to be decisive

1. **Utility collapse in baselines possibly due to hyperparameter choices (R4).**
While exploring alternative hyper-parameterizations could be informative, requiring authors to re-engineer baseline methods may go beyond standard expectations. In this respect, this concern may not be critical.

2. **Lack of formal convergence or optimality guarantees for the min–max formulation (R4).**
While formal convergence or optimality analysis would be highly desirable, the lack of such theory does not constitute a fatal flaw, given the breadth of empirical validation and the focus on robustness-oriented unlearning practice.

3. **Potential new vulnerabilities introduced by the defense (R4).**
While investigating newly introduced vulnerabilities would be valuable, the absence of such findings does not constitute a critical flaw, especially given the broad attack coverage evaluated.

4. **Fairness of comparing two-stage PRISM to single-stage baselines (R4).**
The two-stage design is part of the method itself, and comparisons remain valid as long as inputs and evaluation protocols are consistent. The rebuttal also provides a runtime analysis discussing the trade-off between increased computational cost and the observed utility and robustness gains.

5. **Limited experiments on very large models (70B+) or MoE architectures (R2, R4).**
This is a limitation but not a sufficient reason for rejection, given practical constraints and the demonstrated effectiveness at the 7–8B scale.

6. **Continual unlearning beyond conceptual discussion (R1, R2, R4).**
The authors provide a reasonable conceptual extension, and deeper treatment can be deferred to future work.

### Partially addressed concerns
1. **Computational cost beyond runtime (memory/space overhead) (R1, R2).**
The rebuttal provides detailed runtime analysis, but memory or space overhead is not explicitly reported. This remains a partial limitation, as the parameter-space smoothness component involves additional gradient computations, which can plausibly increase memory usage.
2. **Connection between representation-space smoothness and generalization (R1, R2).**
The added margin analysis supports robustness claims, but the link to broader model generalization is still more intuitive than rigorously demonstrated.
3. **Ablation isolating orthogonal gradient effects between PS and RS (R2).**
While component ablations are provided, experiments explicitly isolating orthogonalized gradients for each smoothness component are limited, leaving some ambiguity in causal attribution.

### Outstanding concerns
1. **Conceptual novelty of “dual-space” smoothness (R4).**
While both smoothness components are individually motivated and empirically effective, the work does not fully establish that their combination yields more than additive benefits, making the “dual-space” framing somewhat weaker than claimed.
2. **Lack of a formal or empirical guarantee that combining both smoothness objectives yields synergistic robustness (R4).**
Although each smoothness component improves robustness to different threat models, the paper does not rigorously show that jointly enforcing both leads to fundamentally stronger robustness than applying them independently.

**Reviewer Scores:**

### Reviewer hH6e (R2).
Reviewer hH6e actively participated in the discussion and explicitly stated that they would maintain their positive assessment after reading the rebuttal. Given that their main concerns were addressed and no new issues were raised during the discussion, **it is reasonable to expect that their original score of 8 would have remained unchanged.**

### Reviewers 3Qb5 (R1) and 6fND (R3).
Neither reviewer participated in the discussion phase. Both reviewers initially assigned borderline scores and did not express strong conviction for acceptance. However, most of their technical concerns—such as margin quantification, probe sensitivity, relearning evaluation, and additional baselines—were addressed in the rebuttal. Under a normal discussion process, **it is reasonable to expect that at least one of the two reviewers might have likely increased their score (e.g., from 4 to 5)**, while the other would likely have maintained their original assessment.

### Reviewer KUGw (R4).
Reviewer KUGw provided a notably critical review and did not engage in post-rebuttal discussion. The authors responded extensively to the raised concerns, including threat models, notation, baselines, computational cost, and robustness analysis. Under a normal discussion process, clarification and exchange could have led to increasing the score. However, given the reviewer’s persistent concerns about conceptual novelty—particularly regarding the “dual-space” framing—it is unlikely that the score would have reached an acceptance level. **A reasonable expectation is a change from 2 to 3**, but not higher.

---

### Decision · Program_Chairs · 2026-01-26

Accept (Poster)